# The Function and Mechanism of Lipid Molecules and Their Roles in The Diagnosis and Prognosis of Breast Cancer

**DOI:** 10.3390/molecules25204864

**Published:** 2020-10-21

**Authors:** Rui Guo, Yu Chen, Heather Borgard, Mayumi Jijiwa, Masaki Nasu, Min He, Youping Deng

**Affiliations:** 1School of Public Health, Guangxi Medical University, 22 Shuangyong Rd, Qingxiu District, Nanning 530021, China; guorui9506@163.com; 2Department of Quantitative Health Sciences, University of Hawaii John A. Burns School of Medicine, 651 Ilalo Street, Honolulu, HI 96813, USA; yuchen8@hawaii.edu (Y.C.); hborgard@hawaii.edu (H.B.); jijiwa@hawaii.edu (M.J.); mnasu@hawaii.edu (M.N.); 3Department of Molecular Biosciences and Bioengineering, College of Tropical Agriculture and Human Resources, University of Hawaii at Manoa,1955 East West Road, Agricultural Sciences, Honolulu, HI 96822, USA

**Keywords:** lipids, breast cancer, lipidomics, diagnosis, prognosis, drug resistance, lncRNAs

## Abstract

Lipids are essential components of cell structure and play important roles in signal transduction between cells and body metabolism. With the continuous development and innovation of lipidomics technology, many studies have shown that the relationship between lipids and cancer is steadily increasing, involving cancer occurrence, proliferation, migration, and apoptosis. Breast cancer has seriously affected the safety and quality of life of human beings worldwide and has become a significant public health problem in modern society, with an especially high incidence among women. Therefore, the issue has inspired scientific researchers to study the link between lipids and breast cancer. This article reviews the research progress of lipidomics, the biological characteristics of lipid molecules, and the relationship between some lipids and cancer drug resistance. Furthermore, this work summarizes the lipid molecules related to breast cancer diagnosis and prognosis, and then it clarifies their impact on the occurrence and development of breast cancer The discussion revolves around the current research hotspot long-chain non-coding RNAs (lncRNAs), summarizes and explains their impact on tumor lipid metabolism, and provides more scientific basis for future cancer research studies.

## 1. Introduction

Breast cancer is a malignant disease that occurs in breast tissue. It occurs more frequently in women. It is also the second leading cause of female mortality. According to the breast cancer report updated by the American Cancer Society in 2019, the incidence of breast cancer in women increased by 0.3% annually from 2012 to 2016 [1]. In contrast, the mortality rate declined 40% from 1989 to 2017, reducing by a total of 375,900 deaths from breast cancer-related diseases. Studies have shown that this result was significantly linked with the widespread promotion of early breast cancer screening [2]. The early diagnosis and proper treatment of breast cancer through early screening is a crucial way to improve the survival rate of patients [3]. Improvements in screening and treatment have significantly reduced breast cancer mortality. Standard clinical screening techniques include mammography [4], ultrasound [5], and magnetic resonance imaging (MRI) [6]. Mammography is not very sensitive to breast tissue that contains a large number of fibrous structures. This type of tissue accounts for most of the examined cases, which causes one-third of missed diagnoses [7,8]. Ultrasound examination is a supplementary assessment to an X-ray examination. Image quality is primarily affected by the skill of the operator, and the examination results are unstable [9]. MRI detection is expensive, not suitable for large-scale early screening, and has a high false-positive rate that makes its detection result unsatisfactory [10,11]. Therefore, it is particularly important to find an early diagnosis method that is easy to operate, affordable, and suitable for detection in a large-sized population.

The evidence from many studies has confirmed that the metabolism of proliferating tumor cells is different from the cells in normal tissues [12]. The occurrence and development of tumors have specific metabolic processes, and the malignant transformation of tumor cell phenotypes mainly occurs in the lipid metabolism process [13]. Previous studies have shown that lipid molecules are closely related to the prognosis and development of diseases, e.g., prostate cancer [14], breast cancer [15], and colorectal cancer [16]. Especially in recent years, with the development of MS technology, it has become more feasible to achieve an early diagnosis for breast cancer patients [17,18], metabolic reprogramming [11,19], cancer staging [20], and a therapeutic intervention response [21]. This technology also provides convenience for obtaining lipid molecules with research value. Generally, MS and NMR spectroscopy techniques are used to identify tumor metabolites [22]. Since MS requires the use of chromatography for the preliminary separation of metabolites, it is usually combined with gas chromatography (GC) or liquid chromatography (LC) [23]. GC/LC–MS and other technologies for the metabolomics analysis of breast cancer patients’ tissues, blood, urine, and other biological tissues are important ways to discover lipid biomarkers and explore new metabolites. 

In this paper, we review the literature on lipid and its metabolites in various experimental models in recent years, and then we summarize the valuable and potential lipid biomarkers to provide the basis for further study on the pathogenesis, diagnostic methods, and adverse clinical prognosis.

## 2. Biological Characteristics of Lipids

Lipids are means as small hydrophobic or amphiphilic molecules that are perhaps derived wholly or partially from the condensation of carbocation groups of thioesters and/or the condensation of carbocation groups of isoprene units [24,25]. Because lipids undergo many different biochemical changes during the biosynthesis process, their structure is more complicated than carbohydrates and proteins. Lipids form a variety of molecules through the combination of different types of building blocks, producing an extremely heterogeneous collection of molecules. In order to unify the classification and naming rules of lipid molecules, the LIPID MAPS Alliance [26] has developed a complete lipid classification system where the lipids are divided into eight categories that include fatty acids (FAs), glycerolipids (GLs), glycerophospholipids (GPs), sphingolipids (SPs), saccharolipids (SLs), polyketides (PKs), sterol lipids (STs), and prenol lipids (PRs). Due to the numerous and complex structures of lipids, we selected representative molecules from each category to represent the structure of lipids (Figure 1).

The chain extension of acetyl Coenzyme A (acetyl-CoA) primer and malonyl-CoA (or methyl malonyl-CoA) group synthesizes FAs. The functional groups are replaced by other atoms to form various molecular groups [27]. Some common FAs are stearic acid, soft acid, and arachidonic acid. GLs are composed of glycerol groups, and the fatty acids also include alkyl and 1Z-alkenyl variants [28]. Another lipid-containing glycerol group is that of GPs. In its molecular structure, the hydroxyl groups at the first and second positions of glycerol are replaced by FAs to form esters. The two fatty acids are nonpolar chains, so they have hydrophobic properties [29]. The hydroxyl group at position three is replaced by a phosphorus-containing group, and the third carbon atom of the phosphate group connected to a glycerol molecule is hydrophilic [30], which leads to GPs being both hydrophilic and hydrophobic. SPs take long-chain nitrogenous bases as their core structures, and they are ubiquitous in eukaryotic membranes, especially in central nervous system tissues. The in-depth study on sphingolipid metabolism and function by an SP reveals its family members, including ceramide (Cer), sphingosine (Sph), sphingosine-1-phosphate (S1P), and ceramide-1-phosphate (C1P) [31]. Unlike the definition of "glycolipids" by the International Union of Pure and Applied Chemistry (IUPAC) [32], an SL is a compound in which the fatty acyl groups are directly connected with a sugar skeleton. STs and PRs share a biosynthetic pathway through the polymerization of dimethylallyl pyrophosphate, but there are vital differences in their final structure and function. A PK is derived from the polymerization of acetyl and propionyl and belongs to the secondary metabolites of animals, plants, and microorganisms [33].

Significant changes in the lipid molecular structure have created a diversity of biological functions [34]. The complex lipid metabolism system is formed by lipid–lipid, lipid–protein, and lipid and other biomolecule interactions [35,36,37]. Lipids, as the medium of cell signal transmission, affect the metabolic behavior of tumor growth, play a role in invasion and metastasis, reshape the metabolic mechanism of cells, and transform the metabolic program from glycolysis to lipid-dependent energy supply [38]. One of the characteristics of cancer is metabolic reprogramming [39,40]. Unlike normal cells, cholesterol in cancer cells mainly exists in the inner layer, and external stimuli change the cholesterol content in the inner and outer layers. Based on this, it has been found that treating cells with a statin can reduce the cholesterol level in the inner layer of a cell membrane, leading to the inhibition of cell growth, which suggests a new method to treat cancer by regulating the cholesterol level of cells with drugs [41]. Secondly, lipids are involved in cell regulation including cell proliferation, apoptosis, and development. Comprehensive lipidomic research of breast has shown that some genes regulating the metabolism of lipids are highly expressed, suggesting that phospholipids perhaps have diagnostic function and the regulation of their metabolism can provide treatment changes for breast cancer [42].

## 3. Research Progress of Lipidomics

In 2003, Han and Gross and others initiated the concept of “lipidomics” [43]. Lipidomics belongs to the category of metabolomics, which is a science that studies the biological characteristics of all lipid molecules in organisms. It has an alternative subject in proteomics and genomics. The formal presentation of the concept means that the study of lipids in organisms has risen from a single level to a systematic level. Therefore, it is possible to achieve efficient and accurate lipidomics research [44]. 

The research content of lipid metabolomics mainly includes four parts: lipid extraction, separation, analysis and detection, and related bioinformatics analysis [45]. Because lipids are insoluble in water, organic solvents are usually used to extract lipids from biological tissues. Commonly used extractants include a methanol–chloroform mixture, methyl tert-butyl ether (MTBE) and solid phase extraction (SPE) [46,47]. The extracted lipid mixture is separated by a chromatographic adsorption method. With the improvement of this separation equipment, scientists have adopted HPLC [48] in recent years to achieve the rapid separation of lipids. However, this separation method is time-consuming, and the UV identification requires π–π * numerical conversion, which cannot achieve a strict proportional analysis with the molecular weight of lipid molecules, resulting in a great reduction in lipid detection accuracy.

High-resolution bio-mass spectrometry is the core tool of lipidomics analysis at present that can close the gap between lipid metabolism and clinical disease manifestations. There are several commonly used methods. The first is electrospray ionization mass spectrometry (ESI) [49]. ESI is a form of mass spectrometry with an electrospray ionization system that has a high sensitivity and is usually used in conjunction with liquid chromatography devices. It is suitable for the analysis of simple FAs and lipids with massive molecular weights. The second is matrix-assisted laser ionization mass spectrometry (MALDI) [50], which is suitable for the blood lipids and highly hydrophobic and nonpolar lipids, (phospholipids, for example). The third method is GC/LC–MS [51], which needs to transform lipid molecules into a gaseous state by derivatization before they can be studied. The most common application is to derivatize a lipid with methanol-forming FA methyl ester (FAME) for downstream analysis. The fourth method is nonaqueous capillary electrophoresis/mass spectrometry (NACE), which has a high separation efficiency. This technique does not require the preparation of a complicated sample, it is relatively low cost, and it is suitable for inactivated fatty acids [52] and phospholipids [53]. The fifth is MALDI imaging [54], which can be utilized to determine the spatial distribution of lipids in tissue samples; it displays the density of various lipid molecules as a heat map and retains their anatomical structures. This method is hypothesis-free and is expected to achieve a higher throughput lipid analysis [55].

Lipidomics, as a new research field, represents a new approach for diagnosis and treatment innovation in the next generation. Mass spectrometry, as a powerful analytical technique, provides convenience for lipidomics to explain disease mechanisms. 

## 4. Lipids and Drug Resistance in Cancers 

Drug action (chemotherapy, targeted drugs, and immune drugs) is an important treatment for cancer patients. However, long term use results in drug resistance that reduces the treatment’s efficacy and accelerates the progress of cancer, ultimately leading to the death of patients. Though the proper combination of anti-cancer drugs has dramatically improved the treatment of malignant tumors, drug resistance to chemotherapy often occurs, and the resistance of cancer to chemotherapy drugs remains the main clinical problem in preventing a recurrence. The ATP-binding cassette (ABC) transporter superfamily is a group of transmembrane proteins. The ABC transporter uses ATP to hydrolyze energy to transport a series of ionic compounds (carbohydrates, lipids, proteins, etc.) into the cell in the opposite direction. In addition, the ABC transporter has the ability to transport heterologous biological substances and drug-conjugated compounds. P-glycoprotein, multidrug resistance-related protein (MRP1), adenosine triphosphate binding transporter G superfamily member 2 antibody (ABCG2), mitoxantrone resistance protein (MXR), and breast cancer resistance proteins 1 (BCRP1) can help the body block the transport of drug molecules, expel anti-cancer drugs from cells, and reduce the effective concentration in cells, thus leading to acquired multidrug resistance. Drugs targeting tumor vessels have a limited effect on the survival rate of patients, which remains attributed to the patient’s resistance to anti-angiogenic drugs (AADs) [56,57]. What is even more surprising is that tumors such as breast cancer, cervical cancer, and gastric cancer growing around adipose tissue have lower benefits [56]. 

The rapid growth and expansion of a tumor depends on large angiogenesis [58]. Blood vessels provide nutrients, hormones, and sugar for cancer cells. The use of AADs reduces angiogenesis and energy supply, which leads to hypoxia in the intracellular environment, triggering fat cells to decompose and store excessive lipids, as well as causing reprogramming of lipid metabolism [38,59]. Further research has shown that the phosphorylation level of an AMP-activated protein kinase in tumor cells increases after AAD treatment, and the increase of carnitine palmitoyl transferase 1 (CTP1) activates fatty acid biosynthetic cells to provide ATP, all of which are caused by hypoxia. CTP1 is the rate-limiting enzyme of FA β-oxidation, which is the core process of fatty acid decomposition. The magical thing is that when CTP1 is knocked down, the drug resistance of hepatic cell carcinoma (HCC) is reversed by using AADs again. Therefore, CTP1 inhibitors can be used to both reduce lipid decomposition and reduce energy support for tumors. Moreover, another energy supply method called the giant cell drinking process has been found to be resistant to drug action [60]. Macrophages take up necessary biological materials from dead tumor cell fragments through endocytosis. Amino acids, sugars, and lipids can be reused to maintain their own expansion. Blocking the process of endocytosis can restore the drug sensitivity of cancer cells. Thus, in many cancer types, the Ras pathway activates downstream effector by stimulating platelet-derived growth factor receptor (PDGFR) and epidermal growth factor receptor (EGFR) [61], e.g., phosphatidylinositol 3-kinase (PI3K), inositol triphosphate (PIP3), and Rac1. PI3K regulates the actin cytoskeleton through Rac1 and promotes the closure of a large body through phosphoinositide signal [62]. The phagocytized necrotic cells are decomposed and metabolized to provide energy for tumor cell proliferation (Figure 2) [38,58,59,60,61,62].

It is widely known that the reduction of glycoproteins and the increase of alpha-fetoproteins and carcinoembryonic antigens on the surface of cancer cells also promote the spread and metastasis of tumor cells. However, the role of lipid molecules in tumor drug resistance is often neglected. In fact, lipid metabolism plays an essential role in regulating tumor sensitivity to anticancer drugs. However, it should be noted that whether it is a CTP1 inhibitor or a macrosomia blocker, it may not be sufficient for cancer treatment alone because this release may not eliminate malignant cells. Clinically, the combination of a cell inhibitor, targeted therapy, and a lipid metabolism inhibitor is more effective in treating cancer [59]. This new mechanism of tumor drug resistance will provide exciting opportunities for the development of new therapies for lipid metabolism in the future.

## 5. Role of Lipid in the Diagnosis of Breast Cancer

Lipids widely exist in human blood, urine, hair, and other tissues with convenient accessibility. Therefore, there have been many valuable lipid molecular screening experiments using lipidomics technology. Experiments have shown that a lipid is a useful biomarker for cancer diagnosis [63,64], especially in breast cancer. Therefore, the authors of this paper summarize the lipid molecules with diagnostic potential related to breast cancer in patients (Table 1) [11,42,65,66,67,68,69,70,71,72,73,74,75,76,77,78,79,80].

In some studies, changes of phospholipids in plasma, serum [74,81], and urine [66,82] of breast cancer patients have been reported. For example, compared with normal tissues, the content of linoleic acid in patients’ tissues is higher. In contrast, the content of oleic and stearic acid is lower [11], especially in tertiary and estrogen receptor (ER)-negative breast cancer [25]. Under the action of the desaturase system, monounsaturated fatty acids introduce unsaturated bonds at precise positions in the fatty chain to form a bend, which changes the biophysical properties of the molecule [83]. The occurrence of tumor cells depends on the fluidity, functionality, and flexibility of monounsaturated fatty acids, which also involves the action of stearoyl-CoA desaturase (SCD-1). Studies have shown that most of the monounsaturated fatty acids in fat are in the form of oleic acid, which is completely obtained through endogenous processes, and the long-term large-scale intake of cis-monounsaturated oleic acid can reduce the risk of breast cancer [84,85]. Linoleic acid is a significant breast cancer cell-stimulating factor. Another finding from population diet control experiments is that linoleic acid has anti-cancer effects [27]. Docosahexaenoic acid (DHA) [75] also has anti-cancer functions. DHA inhibits the growth of breast cancer cells and increases apoptosis [86,87,88]. It induces apoptosis through lipid peroxidation by increasing the level of reactive oxygen species (ROS) and activating the caspase pathway [89]. Lipids such as phosphocholine, choline, and glycerophosphocholine in tumors will also increase, and non-invasive detection by magnetic resonance spectroscopy can be used to separate benign and malignant tumors [90]. Phosphatidylcholine (PC) (32:1) can be used as a diagnostic substance for benign and malignant breast tissue [74], as well as a prognostic marker for triple-negative breast cancer [91]. Serum-free fatty acids (FFAs) are important biologically active molecules. In vitro culture experiments have shown that FFAs have an anti-proliferative effect on breast cancer cells [92]. Chen et al. found that the combination of 15 lipid compounds can be used as both plasma biomarkers to distinguish early cancer and benign lesions and for an auxiliary and mammography examination to avoid the unnecessary follow-up of an incredibly invasive biopsy [11]. Park et al. reported that the four candidate markers (L-octanoylcarnitine, 5-oxoproline, hypoxanthine, and docosahexaenoic acid) found in the plasma of patients can effectively distinguish healthy subjects from malignant breast tumors, and it is expected to reach an early stage by only collecting blood samples [75]. Li et al. found that blood lipids in breast cancer patients after chemotherapy will increase. However, it should be noted that plasma contains a variety of lipoprotein libraries, and each lipoprotein constitutes thousands of different types according to different dietary sources [21]. For example, Meikle et al. pointed out that many phospholipids related to breast cancer have changed after eating soybeans and dairy products [93]. Therefore, when trying to determine the link between blood lipids, breast cancer, and other diseases, it is necessary to consider the impact of dietary intake on the disease.

Changes in the concentration of lipid metabolites reflect changes in the disease state. Monitoring the fluctuations of metabolites in organisms is an innovation in the early detection of breast cancer, but it is still in its infancy. Both the separation and analysis of metabolites require more mature technical and theoretical support. While researching lipid markers, we must also develop new and more valuable lipid molecules to detect early breast cancer.

## 6. Effects of Lipid Metabolism on Proliferation, Migration, and Apoptosis in Breast Cancer

The metabolism of most tumor cells differs from that of healthy cells. In cancer cells, in addition to changes in glucose metabolism, lipid metabolism also undergoes adaptive changes [94,95]. During the rapid proliferation of malignant cells, excess lipids are needed to synthesize biofilms, organelles, and important signal molecules [96,97]. The sources of lipids in the human body include dietary intake and the de novo synthesis of liver cells. The supply of lipids is essential for cell growth, but changes in epigenetics and the local microenvironment of tumors can cause disorders of lipid metabolism and activate cholesterol synthesis [39]. The lipid biosynthesis process can not only provide a large number of necessary bio-membrane components for the rapid division and proliferation of tumor cells but also promote the development of cancer by synthesizing a series of sphingomyelin, phosphatidylinositol, and other oxide signaling molecules [98].

The conserved nuclear protein 54 kD Nuclear RNA-and DNA-binding protein (p54nrb) plays a role in RNA splicing and editing, DNA repair, and gene transcription [99]. These functions are essential for maintaining normal cell growth. When the protein structure is lost or changed, it may lead to the development of cancer [100]. Sterol regulatory element-binding proteins (SREBPs) can activate key enzymes in fatty acid and cholesterol biosynthesis, such as fatty acid synthase (FASN), citrate lyase (ACLY), and acetyl-CoA carboxylase 1 (ACC1). [101]. Some studies have shown that tumor cells can activate the de novo synthesis of fatty acids, which mainly depends on the SREBP transcription factor to increase the expression of adipogenic genes [102]. After experiments, it was determined that p54nrb is a new interacting protein of SREBP-1a [103]. SREBP-1a binds to p54nrb through residue Y267 to control target gene expression and lipid synthesis in breast cancer cells. In the process of fatty acid synthesis, various enzymes play a critical regulatory role. Since the discovery of FASN, its role in tumor growth and intracellular signal transmission has been extensively studied. Further studies have found that the use of FASN drug inhibitors can cause breast cancer cells to undergo apoptosis or tumor cell shrinkage, which indicates that FASN has a protective effect on breast cancer cells [104,105,106]. The reason for this is that the inhibition of fatty acid synthase leads to the aggregation of malonyl-CoA, which increases the expression of ceramide. It induces the pro-apoptotic gene bcl-2 and adenovirus E1B 19 kDa-interacting protein 3 (BNIP3), tumor necrosis factor-related apoptosis-inducing ligand (TRAIL), and death-related protein kinase 2 (DAPK2), thus significantly up-regulating and promoting cell apoptosis [105]. In addition, ACLY plays an important role in fatty acid biosynthesis and is the main enzyme-producing acetyl-CoA. A study has shown that microRNA-206 can encode the cell membrane surface integrin receptor CD49b, which acts on ACLY to cause breast cancer cell metastasis [107] (Figure 3). 

In addition, the development and metastasis of cancer are also closely related to the imbalance of the body’s redox balance [108]. This imbalance has been shown to be caused by an increase in free radicals (mainly ROS) [109]. ROS are a byproduct produced during aerobic metabolism [110]. They react with polyunsaturated fatty acids in biomembrane phospholipids to trigger lipid peroxidation and generate a series of complex products such as oxygen free radicals under peroxidation conditions. Many studies have shown that lipid peroxidation products may be involved in inducing cell apoptosis [111] because lipid peroxidation mainly acts on biological membranes and mitochondria-containing DNA. As the engine of the body, mitochondria participate in the physiological process of diseases such as aging, new blood vessels, and cancer. Its damage has a significant impact on the organism [112]. In addition, the role of ROS in tumorigenesis and development is also complicated [113], triggering various effects at different concentrations. On the one hand, accumulated ROS can cause irreversible oxidative damage to lipids, proteins, and DNA, and they can activate the ROS-mediated NF-κB pathway to cause chronic inflammation [31,114]; on the other hand, it can regulate intracellular ROS level to kill tumor cells. ROS open the mitochondrial permeability transition pore (PTP), increases the permeability of the inner membrane, interferes with the mitochondrial membrane potential, and induces the release of cytochrome c. Cytochrome c, apoptosis protease activator 1, and caspase 9 form an "apoptotic body" that initiates changes in the downstream process of the apoptosis cascade [109,115]. Therefore, targeted research on the dual effects of ROS is a potential direction for the treatment of malignant tumors.

## 7. Lipids as Predictors of Breast Cancer Risk and Prognosis

For breast cancer patients, lymph node status is the most accurate and important survival predictor. In addition, tumor size, the degree of differentiation, and histological type can also be used as prognostic factors. With the in-depth study of lipid molecules, people have discovered changes in lipid levels in the prognosis of different types of cancer. This abnormal change is associated with the survival risk of a variety of cancers, including breast cancer [116,117,118]. The negative correlation between lipid levels and the poor prognosis of cancer seems to be due to tumor type and the characteristics and pathophysiological effects of lipid molecules.

Triglycerides (TGs) and cholesterol (CHO) play important functions in biofilm composition and energy storage; 27-hydroxycholesterol (27HC) is the main metabolite of high cholesterol and is also a ligand for the estrogen receptor and liver X receptor (LXR) [119]. It is composed of sterol 27 hydroxylase (CYP27A1) and cytochrome P450 oxidase. Compared with normal tissues, the expression of 27HC in malignant breast tissues is significantly higher, and the increase of synthase CYP27A1 can improve the tumor grade [120]. When the production of CYP27A1 is cut off, it can significantly reduce the distant metastasis of tumor cell-related models [121]. 27HC can regulate the activity of estrogen receptor alpha (ERα)-positive breast cancer and directly act on immune cells to affect tumor growth and metastasis [122]. In addition, estrogen produced by adipose tissue also increases the risk of ERα-positive breast cancer. A number of epidemiological investigations and studies have found that the estrogen and progesterone levels of postmenopausal women increase the incidence and mortality of breast cancer [123,124]. A retrospective investigation on the relationship between blood lipid levels and breast cancer in patients found that low preoperative TG and high-density lipoprotein cholesterol (HDL-C) levels are risk factors. They may be independent predictors in the prognosis of breast cancer [125]. It is hypothesized that TGs cause a decrease in the sex hormone binding globulin and an increase in free estradiol, thereby increasing the risk of breast cancer recurrence. Nevertheless, another study showed that after excluding the confounding effects of insulin and body mass index, there was a failure to prove that blood lipids (TG and low-density lipoprotein cholesterol (LDL-C)) levels and breast cancer prognosis have any statistically significant correlation [126]. Blood lipid levels are potentially affected by menopausal status and fat intake. HDL-C can prevent lipid peroxidation by inhibiting the oxidative damage produced by LDL-C [127]. HDL-C is related to the incidence of breast cancer in postmenopausal women and is regarded as a marker of androgenic status [128]. When the body ingests a large amount of fat for a short time, it stimulates the increase of circulating estrogen, causing cell damage and affecting the growth of cancer cells. A Mendelian randomized grouping experiment proved that genetically elevated LDL-C due to proprotein convertase subtilisin/kexin type 9 (PCSK9) gene mutations could increase the incidence of estrogen receptor-positive breast cancer. The use of statins to target and repair mutant genes resulted in a finding that the mutation effect is not apparent [129]. 

Triple-negative breast cancer (TNBC) is one of the subtypes of breast cancer with a high risk of early recurrence and a low survival rate. A study found that the content of PC (32:1) in recurrent TNBC was significantly higher than that in non-recurrent TNBC. Compared with estrogen receptor-positive and human epidermal growthfactor receptor 2 (HER2) breast cancer, TNBC contained more PC (32:1) and PC (30:0) [130], suggesting the potential of phospholipids as candidate predictors [42,91]. In addition, PC is also considered to be a lipid metabolite that promotes cancer metastasis and malignant transformation [131]. On the other hand, dyslipidemia caused by high-fat diet and obesity can increase women’s risk of disease and adversely affect the prognosis of breast cancer patients [132,133]. An exogenous diet consumes much fat. Breast cancer cells absorb fat particles floating in the blood to promote their proliferation. This is an unexpected fat particle absorption mechanism that recently discovered and that revealed the connection between cancer cells and dietary fat [134]. Lipid metabolism is interfered by both exogenous and endogenous processes, and the mechanism of influence on breast cancer risk and prognosis is still unclear, so further prospective studies and experimental proof are needed.

## 8. LncRNA Related to Lipid Metabolism in Cancer

Long-chain non-coding RNAs (lncRNAs) have shown immeasurable potential value as a hotspot in molecular research, especially in the rapid development of tumor metabolic reprogramming. Some studies have shown that lncRNA can affect cancer migration, invasion, and other pathological processes [135]. 

lncRNAs are defined as transcripts of more than 200 nucleotides in length without a coding ability [136]. LncRNAs can participate in the regulation of a variety of lipid metabolism-related genes. Hepatocellular carcinoma up-regulated long non-coding RNA (HULC) [137,138] was the first reported liver-specific lncRNA that is highly expressed in liver cancer. HULC induces methylation at the CpG site at the miR-9 promoter. The methylation level of miR-9-1 is highly correlated with the increased risk of recurrence, resulting in a significantly shorter survival time for clear cell renal cell carcinoma [139]. HULC has been found to eliminate the miR-9-mediated inactivation of the transcription factor PPARA (proliferator-activated receptor alpha), up-regulated the same transcription factor [140,141], and activated liver cancer cell Acyl-CoA synthetase long-chain1 (ACSL1) [142,143], which promotes the expression of triglycerides and cholesterol accumulation in cells. According to reports, long non-coding hepatitis C virus (HCV) regulated 1 (lncHR1) is involved in tumor lipid metabolism, and the expression of SREBP-1c has been maintained by regulating the phosphorylation level of PDK1 upstream of AKT. SREBP-1c is an important regulator that regulates the dynamic balance of lipid metabolism and encodes the transcription of a variety of enzymes related to the synthesis of TG and FA to accelerate lipid synthesis. In addition, lncRNA related to cervical cancer lymph node metastasis promotes fatty acid metabolism reprogramming by regulating fatty acid binding protein 5 (FABP5) and promoting cervical cancer lymph node metastasis [144] (Table 2).

In short, lncRNAs participate in lipid metabolism in different ways to affect tumor progression. We have explored the effect of lncRNAs on lipid metabolism and understand the mechanism of tumor carcinogenesis. These all provide new ideas and strategies for tumor treatment.

## 9. Conclusions

Lipids play an important role in the occurrence and development of cancer. They transmit information through communication between cells, regulate cell metabolism, and maintain an internal stable state. Studies have shown that lipid molecules are potential biomarkers for biological diagnosis. Combined with mass spectrometry as an auxiliary detection tool for mammography, they are expected to realize non-invasive early screening of breast cancer. In addition, with the continuous development of lipidomics technology, combining lipidomics with other omics technologies can give full play to the role of lipidomics in understanding the molecular mechanisms of disease. Though there have been many studies on the relationship between lipids and various diseases at home and abroad, there have been few studies on how lipids and their metabolism are involved in the occurrence and development of breast cancer. This is caused by both the structural diversity of lipid types and the immaturity of separation technology, which are challenges that need to be overcome in lipidomics research. Therefore, it is necessary to combine the existing biological analysis technology and emerging interdisciplinary theory to make a breakthrough in the research of lipidomics.

## Figures and Tables

**Figure 1 molecules-25-04864-f001:**
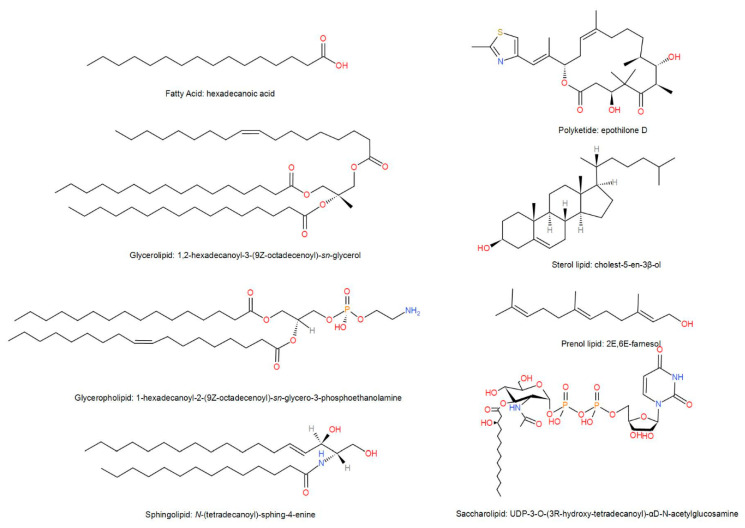
Representative structure of lipids.

**Figure 2 molecules-25-04864-f002:**
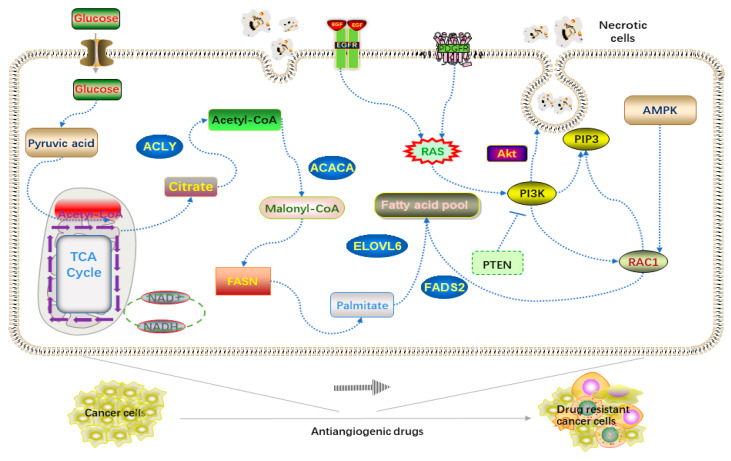
Mechanism of drug resistance of tumor cells induced by anti-angiogenic drugs (AADs).

**Figure 3 molecules-25-04864-f003:**
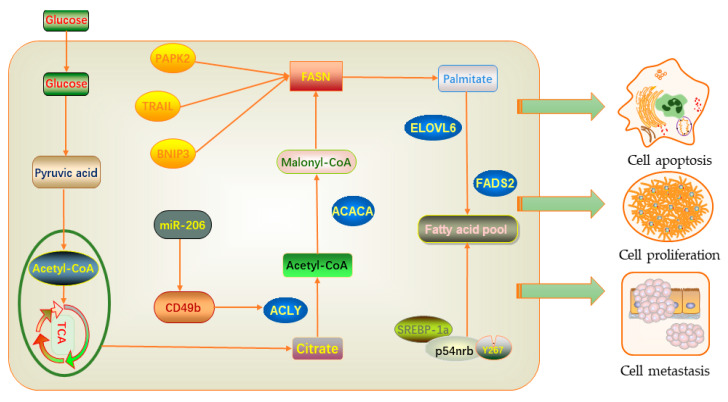
Lipid-related regulatory factors promoting the occurrence and development of breast cancer.

**Table 1 molecules-25-04864-t001:** Lipids with the ability of early diagnosis in breast cancer.

Diagnosis Biomarker	Sample Type	Lipidomics Technique	Reference
PCs, ePC (38:3) and ePC (32:2), PC ae C42:5 and PC aa C42:2	Urine, Plasma, Tissue	nLC-ESI-MS-MS, ESI-MS/MS, UPLC-MS	[11,42,65,66]
PC (32:1), (34:4), (38:3), (40:5), (40:3), (44:11)	Plasma, Tissue	ESI-MS/MS, MALDI-IMS	[11,67]
PC (30:0), (34:0), (34:1), (36:0), (36:1), (36:2), (38:4), (40:6), (42:6)	Tissue	MALDI-IMS, probe electrospray ionization-MS	[67,68]
PC (20:2/20:5) and PC (20:0/24:1;)	Plasma	UPLC-quadrupole time-of-flight tandem/MS	[69]
LPC (18:0), (18:3), (20:0), (20:1), (20:2) and a C (16:0)	Plasma, Urine, Tissue	ESI-MS/MS, MALDI-IMS, nLC-ESI-MS-MS	[11,65,67]
PEs and PE (15:0/19:1)	Plasma, Tissue, Urine	UPLC-quadrupole time-of-flight tandem/MS, HILIC-HPLC/ ESI-MS, nLC-ESI-MS-MS	[66,69,70]
Triglyceride, Triglyceride (12:0/14:1)	Plasma	UPLC-quadrupole time-of-flight tandem/MS	[69,71,72,73]
Diglyceride (18:1/18:2)	Plasma	UPLC-quadrupole time-of-flight tandem/MS	[69]
C19:0 CE, C19:1 CE, C19:2 CE	Plasma	ESI-MS/MS	[11]
PI (16L:0/16:1) and PI (18:0/20:4)	Plasma	normal-phase/reversed-phase two-dimensional LC-MS	[74]
Docosahexaenoic acid	Plasma, Tissue	LC-MS, Raman spectroscopy	[75,76]
Fatty acids: C14:0, C16:0, C16:1, C18:0, C18:3, C18:2, C20:4, and C22:6	Serum	Chip-based direct-infusion nano-electrospray ionization-Fourier transform ion cyclotron resonance mass spectrometry, GC–MS	[76,77,78]
HDL-C, VLDL-C, LDL-C, TC	Plasma	NA	[72,79,80]
N-palmitoyl proline	Plasma	UPLC-quadrupole time-of-flight tandem/MS	[69]

PC: phosphatidylcholine; ePC: ether-linked phosphatidylcholine; LPC: lysophosphatidylcholine; PE: phosphatidylethanolamine; CE: cholesteryl esters; PI: phosphatidylinositol; NA: not applicable; LDL-C: low-density lipoprotein cholesterol; HDL-C: high-density lipoprotein cholesterol; VLDL-C: very low-density lipoprotein cholesterol; TC: total cholesterol; UPLC: ultra-HPLC; IMS: imaging mass spectrometry; HILIC: hydrophilic interaction liquid chromatography.

**Table 2 molecules-25-04864-t002:** Long-chain non-coding RNAs (lncRNAs) related to lipid metabolism in cancer.

LncRNAs	Lipid Metabolism-Related Genes	Cancer Types	Influences	References
HC	hnRNPA2B1	Hepatic carcinoma	TG and CHO decreased	[145]
HAGLROS	FASN, ACC, SCD1, CPT1, SREBP1, PPARγ	Intrahepatic cholangiocarcinoma	Promote autophagy, improving lipid metabolism reprogramming	[146]
MACC1-AS1	ACS, CPT1A	Gastric cancer	Drug resistance	[147]
HULC	ACSL1, PPARA	Hepatic carcinoma	TG and CHO accumulated	[148]
LNMICC	FABP5	Cervical cancer	Lymph node metastasis	[149]
LeXis	RALY	Hepatic carcinoma	CHO increased	[150]
HR1	SREBP-1c, FASN	Hepatic carcinoma	TG and lipid droplet accumulated	[151]
HCP5	CPT1	Gastric cancer	Drug resistance	[152]
MALAT1	SREBP-1c	Hepatic carcinoma	Hepatic steatosis and insulin resistance	[153]

ACC: acetyl-CoA carboxylase; SCD1: stearoyl-CoA desaturase1; CPT1: carnitine palmitoyl transferase 1; ACS: acetyl-coenzyme A synthetase; RALY: heterogeneous ribonucleoprotein; CHO: cholesterol; TG: triglyceride; PPARA: proliferator-activated receptor alpha; FASN: fatty acid synthase; HULC: hepatocellular carcinoma up-regulated long non-coding RNA; SREBP: Sterol regulatory element-binding protein; FABP: fatty acid binding protein 5.

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
