# Peer review of "The Function and Mechanism of Lipid Molecules and Their Roles in The Diagnosis and Prognosis of Breast Cancer"

_molecules, 2020, doi:10.3390/molecules25204864_

Round 1

Reviewer 1 Report

The manuscript entitled "The Function and Mechanism of Lipid Molecules and Their Roles in The Diagnosis and Prognosis of Breast Cancer" "by Guo et al., describes the research progress of lipidomics and the relationship between some lipid molecules and cancer drug resistance. Moreover, the authors discussed the role of specific lipids and lipidomics approaches, mainly the high-resolution bio-mass spectrometry, providing important evidences on the diagnosis and prognosis of breast cancer.

The review is interesting and technically the manuscript is well written. The following points should be addressed to ameliorate the contents:

  1. a figure based on the general structures could help, highlighting the differences of lipid classes.
  2. The authors discussed the role of the membrane lipids in multidrug resistance. However, emphasis is not given on the biophysical membrane properties and the role of balance between the passive cellular influx and active efflux of drug molecules. The well-known alterations of membrane lipid composition of cancer cells must be mentioned.
  3. As increased ROS level can inhibit tumor cell growth, the authors must also stress the free radical involvement in cancer progression and treatment. Apart from the relationship between oxidative stress and cancer progression, the important role of lipid damage associated to lipid peroxidation of polyunsaturated fatty acids must be presented.
  4. Emphasis should be given on the role of monounsaturated fatty acids and desaturase enzymatic activities. Recently I crossed the following publication that might be of interest to the Authors (Metabolites, 2020, 10(9):E345. doi:10.3390/metabo10090345)
  5. A reorganization of the Discussion part is needed. Long-chain non-coding RNAs (lncRNAs) can be presented in a separate section. Discussion can combine aspects of biophysics and lipid composition of cell membranes, cellular proliferation and metabolic changes in breast cancer, as previously suggested.
  6. Also Concluding remarks are necessary.

Reviewer 2 Report

In this manuscript titled "the function and mechanism of lipid molecules and their roles in the diagnosis and prognosis of breast cancer", the authors systematically summarized the progression of lipid study in cancers, especially in breast cancer. They discussed the relationship between some lipids and cancer drug resistance. In general, this review is well writing and comprehensive.  Just two minor comments:  1) the "research progress of lipidomics" part, the authors discussed some techniques on lipidomics analysis, this is not so relevant to this topic. 2) For breast cancer, the hormones like estrogen and 27-hydroxycholesterol(27HC) are very essential for breast cancer development. If the authors could discuss a little about cholesterol metabolites in this review that will make the paper more comprehensive. 

Author Response

Thank you for your letter and for the reviewer’s comments concerning our manuscript entitled “The Function and Mechanism of Lipid Molecules and Their Roles in the Diagnosis and Prognosis of Breast Cancer” (ID: molecules-934687). Those comments are all valuable and very helpful for revising and improving our paper, as well as adding important guiding significance to our research. We have studied the comments carefully and have made correction which we hope meet with approval. Revised portions are marked in red in the paper. The main corrections in the paper and the response to the reviewer’s comments are as follows.

Response to Reviewer Comments

Point 1: The "research progress of lipidomics" part, the authors discussed some techniques on lipidomics analysis, this is not so relevant to this topic.

Response 1: Thank you very much for your suggestion. After careful discussion, we decided to keep this part of the content. The development of lipidomics has promoted the study of lipid functions. The development of mass spectrometry technology has promoted the large-scale study of cell lipids in lipidomics, and realized the high-resolution, high-sensitivity and high-throughput analysis of various lipids, especially phospholipids. The introduction of lipidomics technology is added to this review, which will help readers to fully understand the development process of lipidomics and provide basic theoretical knowledge for the continuous advancement of lipidomics technology in the future.

Point 2: For breast cancer, hormones like estrogen and 27-hydroxycholesterol(27HC) are very essential for breast cancer development. If the authors could discuss a little about cholesterol metabolites in this review that will make the paper more comprehensive.

Response 2: Thank you very much for your suggestion. Cholesterol metabolites are indispensable for the study of the pathogenesis of breast cancer. We ignored the description of metabolites such as estrogen and 27-hydroxycholesterol, so the following content was added to the comment to make the article more comprehensive.

“Triglycerides (TG) and cholesterol (CHO) play important functions in biofilm composition and energy storage. 27 Hydroxycholesterol (27HC) is the main metabolite of high cholesterol and is also a ligand for the estrogen receptor and liver X receptor (LXR). It is composed of sterol 27 hydroxylase (CYP27A1) and cytochrome P450 oxidase. Compared with normal tissues, the expression of 27HC in malignant breast tissues is significantly higher, and the increase of synthase CYP27A1 can improve tumor grade. When the production of CYP27A1 is cut off, it can significantly reduce the distant metastasis of tumor cell-related models. 27HC can regulate the activity of estrogen receptor alpha (ERα)-positive breast cancer, and directly act on immune cells to affect tumor growth and metastasis. In addition, estrogen produced by adipose tissue also increases the risk of ERα-positive breast cancer. The number of epidemiological investigations and studies have found that the estrogen and progesterone levels of postmenopausal women increase the incidence and mortality of breast cancer.”

Reviewer 3 Report

The authors, in this review, summarizes previously published studies on “the function and mechanism of lipid molecules and  their roles in the diagnosis and prognosis of breast  cancer”

Although the topic is interesting, many recent published discovery on lipids and breast cancer have been missed from the authors (for example 2019-2020) .

Furthermore, the discussion should focus on breast cancer and not on other types of cancer and they should add the conclusions of what they describe in their review.

Author Response

Thank you for your letter and for the reviewer’s comments concerning our manuscript entitled “The Function and Mechanism of Lipid Molecules and Their Roles in the Diagnosis and Prognosis of Breast Cancer” (ID: molecules-934687). Those comments are all valuable and very helpful for revising and improving our paper, as well as adding important guiding significance to our research. We have studied the comments carefully and have made correction which we hope meet with approval. Revised portion are marked in red in the paper. The main corrections in the paper and the response to the reviewer’s comments are as follows.

Response to Reviewer Comments:

Point 1: Although the topic is interesting, many recent published discoveries on lipids and breast cancer has been missed from the authors (for example 2019-2020).

Response 1: Thank you very much for your suggestion. We once again collected some of the latest literature on lipids and breast cancer and applied these literatures to the review.

  1. Jasbi, P., et al., Breast cancer detection using targeted plasma metabolomics. J Chromatogr B Analyt Technol Biomed Life Sci, 2019. 1105: p. 26-37.
  2. Gandhi, N. and G. Das, Metabolic Reprogramming in Breast Cancer and Its Therapeutic Implications. Cells, 2019. 8(2).
  3. Azab, S., R. Ly, and P. Britz-McKibbin, Robust Method for High-Throughput Screening of Fatty Acids by Multisegment Injection-Nonaqueous Capillary Electrophoresis-Mass Spectrometry with Stringent Quality Control. Anal Chem, 2019. 91(3): p. 2329-2336.
  4. Wu, Z., et al., "Lipidomics": Mass spectrometric and chemometric analyses of lipids. Adv Drug Deliv Rev, 2020.
  5. Zeleznik, O.A., et al., Circulating Lysophosphatidylcholines, Phosphatidylcholines, Ceramides, and Sphingomyelins and Ovarian Cancer Risk: A 23-Year Prospective Study. J Natl Cancer Inst, 2020. 112(6): p. 628-636.
  6. Iwano, T., et al., Breast cancer diagnosis based on lipid profiling by probe electrospray ionization mass spectrometry. Br J Surg, 2020. 107(6): p. 632-635.
  7. Park, J., et al., Plasma metabolites as possible biomarkers for diagnosis of breast cancer. PLoS One, 2019. 14(12): p. e0225129
  8. Adorno-Cruz, V., et al., ITGA2 promotes expression of ACLY and CCND1 in enhancing breast cancer stemness and metastasis. Genes & Diseases, 2020.
  9. Lupien, L.E., et al., Endocytosis of very low-density lipoproteins: an unexpected mechanism for lipid acquisition by breast cancer cells. J Lipid Res, 2020. 61(2): p. 205-218.
  10. Seyed Hosseyni, M., et al., Evaluation of Expression Levels of Linc-ROR and HULC Genes in Breast Cancer Cells (MCF7) Following Treatment with Nanocurcumin. Journal of Human Genetics and Genomics, 2020. 3(1).
  11. Ferreri, C., et al., Fatty Acids and Membrane Lipidomics in Oncology: A Cross-Road of Nutritional, Signaling and Metabolic Pathways. Metabolites, 2020. 10(9).

Point 2: Furthermore, the discussion should focus on breast cancer and not on other types of cancer and they should add the conclusions of what they describe in their review.

Response 2: Thank you very much for your suggestion. This is a good idea. The conclusion of the article is very necessary. We added a conclusion at the end of the article. This can be used as a summary of the article and help readers better understand the content of the article. The following is our conclusion.

“Lipids play an important role in the occurrence and development of cancer. They transmit information through communication between cells, regulate cell metabolism and maintain an internal stable state. Studies have shown that lipid molecules are potential biomarkers for biological diagnosis. Combined with mass spectrometry as an auxiliary detection tool for mammography, it is expected to realize non-invasive early screening of breast cancer. In addition, with the continuous development of lipidomics technology, combining lipidomics with other omics technologies can give full play to the role of lipidomics in understanding the molecular mechanisms of diseases. Although there are many studies on the relationship between lipids and various diseases at home and abroad, there are few studies on how lipids and their metabolism are related to the occurrence and development of breast cancer. This is caused by both the structural diversity of lipid types and the immaturity of separation technology, which are challenges that need to be overcome in lipidomics research. Therefore, it is necessary to combine existing biological analysis techniques with emerging interdisciplinary theories to make breakthroughs in lipidomics research.”